# Study on the Application of Nitrogen-Doped Holey Graphene in Supercapacitors with Organic Electrolyte

**DOI:** 10.3390/nano13101640

**Published:** 2023-05-14

**Authors:** Yu-Ren Huang, Nen-Wen Pu, Guan-Min Wu, Yih-Ming Liu, Ming-Hsien Lin, Yi-Le Kwong, Siou-Cheng Li, Jeng-Kuei Chang, Ming-Der Ger

**Affiliations:** 1Department of Applied Science, R.O.C. Naval Academy, Zuoying, Kaohsiung 813, Taiwan; g981101@gmail.com; 2Department of Electrical Engineering, Yuan Ze University, Zhongli, Taoyuan 320, Taiwan; 3Department of Chemical & Materials Engineering, Chung Cheng Institute of Technology, National Defense University, Dasi, Taoyuan 335, Taiwan; takululiu@gmail.com (Y.-M.L.); mslin479@gmail.com (M.-H.L.); 4Department of Materials Science and Engineering, National Yang Ming Chiao Tung University, Hsinchu 30010, Taiwan; jkchang@nycu.edu.tw

**Keywords:** holey graphene, activated carbon, N-doping, organic electrolyte, supercapacitor

## Abstract

We present a facile low-cost method to produce nitrogen-doped holey graphene (N-HGE) and its application to supercapacitors. A composite of N-HGE and activated carbon (AC) was used as the electrode active material in organic-electrolyte supercapacitors, and the performances were evaluated. Melamine was mixed into graphite oxide (GO) as the N source, and an ultra-rapid heating method was used to create numerous holes during the reduction process of GO. X-ray photoelectron spectra confirmed the successful doping with 2.9–4.5 at.% of nitrogen on all samples. Scanning electron micrographs and Raman spectra revealed that a higher heating rate resulted in more holes and defects on the reduced graphene sheets. An extra annealing step at 1000 °C for 1 h was carried out to further eliminate residual oxygen functional groups, which are undesirable in the organic electrolyte system. Compared to the low-heating-rate counterpart (N-GE-15), N-HGE boosted the specific capacity of the supercapacitor by 42 and 22% at current densities of 0.5 and 20 A/g, respectively. The effects of annealing time (0.5, 1, and 2 h) at 1000 °C were also studied. Longer annealing time resulted in higher capacitance values at all current densities due to the minimized oxygen content. Volumetric specific capacitances of 49 and 24 F/cm^3^ were achieved at current densities of 0.5 and 20 A/g, respectively. For the high-power-density operation at 31,000 W/kg (or 10,000 W/L), an energy density as high as 11 Wh/kg (or 3.5 Wh/L) was achieved. The results indicated that N-HGE not only improved the conductivity of the composite supercapacitors but also accelerated ion transport by way of shortened diffusion paths through the numerous holes all over the graphene sheets.

## 1. Introduction

In recent years, supercapacitors (SCs), which bridge the gap between rechargeable batteries and traditional capacitors, have attracted great interest of researchers due to their excellent performances, such as high power density, long-cycle stability, rapid charge and discharge rate, and low cost [1,2,3,4]. In particular, electric double-layer capacitors (EDLCs), which store the charge at the electrode-electrolyte interface using non-Faraday charge separation, are promising energy-storage devices because of their fast speed and superior cycle life [5,6]. The capacitance of SCs is directly proportional to the specific surface area (SSA) of the electrode material, so nano-carbon materials with high SSA are widely used. For example, activated carbon (AC), which is very low-cost and possesses extremely large SSA (up to 3000 m^2^/g), has become the most commonly used electrode material [7,8]. However, the low conductivity and inefficient ion transport (due to the small pore sizes and the long diffusion paths) of AC hinders its development because of the poor high-rate performances [9].

Despite the slightly smaller SSA (usually several hundred m^2^/g) of graphene compared to AC, it has many excellent properties, such as high electrical conductivity, thermal conductivity, chemical stability and mechanical strength, which have made them the focus of many EDLC research activities [10,11,12,13,14,15,16,17]. However, well-exfoliated graphene has a much lower packing density in the electrode than AC, which is detrimental for applications requiring high volumetric capacitance [18]. Additionally, the restacking tendency of graphene sheets during the preparation of electrode not only reduces the accessible SSA but also hinders the ions diffusing in and out of the interlayer space [18,19,20,21]. 

Three-dimensional (3D) hierarchical porous structures have been utilized by several researchers to overcome the restacking problem. For example, Kim et al. [22] demonstrated the fabrication of highly porous graphene-derived carbons with hierarchical pore structures in which mesopores are integrated into macroporous scaffolds. The macropores were created by assembling graphene hollow spheres, and the mesopores were derived from KOH activation. Their material exhibited a high gravimetric specific capacitance of 174 F/g in an ionic liquid electrolyte. Tran et al. [23] also reported a method to prepare a graphene/*α*-MnO_2_ nanowire hydrogel via a simple hydrothermal route, in which graphene and *α*-MnO_2_ nanowires are self-assembled into 3D macroporous network structures. Using an asymmetric supercapacitor structure, they achieved a voltage window of 2.0 V and a specific capacitance of 274 F/g. In 2020, Tran et al. [24] developed an ingenious 3D printing technique to form graphene aerogels with 3D hierarchical structure using a specially designed graphene nanoinks. The specific capacitance of their graphene aerogels was ~77 F/g at a current density of 1 A/g.

Modifying the morphology of graphene to further improve its electrochemical performance has also been reported by many researchers. Corrugated graphene [19], curved graphene [20], and crumpled graphene balls [25] have been demonstrated to improve the capacitance because these structures can effectively prevent the problem of graphene restacking, which significantly deteriorate the capacitor performances. In 2015, Peng et al. proposed a simple method to produce holey graphene [21], which offered an excellent capacitance (350 F/g at 0.1 A/g) and high-rate capability (170 F/g at 50 A/g) due to the much shorter ion diffusion paths via the numerous holes as well as the anti-restacking rough surface morphology. However, they used an aqueous electrolyte, whose potential window Δ*V* is smaller than those of organic electrolytes (~2.5–3.0 V). Note that the energy density is proportional to (Δ*V*)^2^. Xu et al. [11] fabricated a supercapacitor using a holey graphene framework (HGF), which were formed by conjugating holey graphene sheets into a free-standing 3D network. Their electrode was binder-free and delivered a specific capacitance of 298 F/g in organic electrolyte. Both the holey graphene and various 3D porous graphene structures can effectively improve the electrochemical performances of SCs due to the significantly shortened ion diffusion paths through the mesopores/macropores or the through-holes. Usually, 3D porous graphene structures offer better anti-stacking capabilities and better specific capacitances.

Another important limitation for commercial applications is the low tap density and mass loading for the graphene electrode material, which result in low volumetric capacitance (F/cm^3^) despite its high gravimetric capacitance (F/g). To increase the volumetric capacitance and to extend the Δ*V*, Huang el al. [18] used an organic electrolyte with a potential window of 2.5 V, and incorporated activated carbon (AC) into holey graphene (10:1, by weight) to produce a composite electrode material with a higher tap density. They achieved high volumetric capacitance (10.3 F/cm^3^) at high rate charging/discharging (20 A/g).

Doping with heteroatoms is another approach for tuning and controlling the electronic properties of graphene [26,27,28,29]. The specific capacitance and the high-rate charge/discharge capability of graphene-based electrode can be improved by n- or p-type doping, which may increase the conductivity of graphene and their binding energies with ions in the electrolyte [30,31]. In 2011, Jeong et al. doped graphene with nitrogen atoms using a plasma process [30], and achieved a capacitance of ~280 F/g (at low current density), which was about four times the value of pristine graphene. 

In 2017, Śliwak et al. synthesized N-doped graphene by a hydrothermal route using amitrole as the nitrogen source [31], and achieved very high N contents of 10.9–13.4 at.%. A high capacitance of 209 F/g (in contrast to 142 F/g for the undoped counterpart) was obtained at 20 A/g in 6 M KOH aqueous electrolyte, and the rate capability was as high as 87% from 0.2 to 20 A/g. They mentioned that the N-atoms in graphene increases the wettability in the electrolytes. Moreover, quaternary nitrogen groups improves the carrier density and hence the conductivity of the graphene due to the contribution of the extra p-electrons to the graphene π-system.

Here we combined the techniques of forming holey morphology and nitrogen doping on graphene in order to maximize the performances of SCs. Moreover, organic electrolyte was chosen to widen the operating potential window, thereby increasing the specific energy. Additionally, to solve the problem of low volumetric capacitance for graphene, a composite of AC and graphene was used as the electrode material. 

## 2. Experimental Section

### 2.1. Preparation of N-Doped Graphene Samples

Natural graphite (Alfa Aesar; particle size: ~70 μm; purity: 99.999%) was first chemically oxidized into graphite oxide (GO) using a modified Staudenmaier method [32]. To prepare N-doped holey graphene, melamine (MA) was chosen as the N source. Amounts of 1 g of GO powder and 0.2 g of MA were added into 100 mL of deionized water (DI), and the mixture was ultrasonicated for 10 min for homogenization. After further stirring for 30 min, the solution was freeze-dried, and then ground into powder (MA-GO). To thermally reduce MA-GO, the powder was put in an alumina boat and heated in a tube furnace at different rates (15 °C/min or 60 °C/min) from room temperature to 600 °C. The reduced N-doped graphene samples are named N-GE-15 and N-GE-60. To obtain N-doped holey graphene (N-HGE), an ultra-rapid heating method was used: a long-handled stainless-steel spoon containing the MA-GO powder was rapidly inserted into the center of the quartz tube (preheated to 600 °C) and dumped onto the tube surface. Upon contacting the quartz tube, it immediately (in about 2 s) reduced/exfoliated into N-HGE. In this way, an extremely high heating rate was realized, and thus the CO_2_ gas generated between atomic layers built up a huge pressure, which not only exfoliated layers but also punched many holes on each graphene sheet. Finally, the furnace was filled with Ar and H_2_ gases (Ar:H_2_ = 5:1), and then heated to 1000 °C and held for 1 h to further reduce the remaining oxygen functional groups. For N-HGE, the duration of the 1000 °C annealing was also varied (0.5, 1, and 2 h) in order to study the effect of residual oxygen functional groups on the performance of SCs. Concerning the yields of the prepared materials, the major material loss occurs in the two heating processes converting MA-GO into N-HGE. First, in the ultra-rapid heating step, typically a weight loss of about 60% occurs; second, during the 0.5 to 2 h of 1000 °C annealing process, the weight loss ranges from 25% to 50%. So the overall yield for N-HGE is around 20–30%.

### 2.2. Characterization

A scanning electron microscope (SEM, Hitachi S-4800) was used to examine the microstructure of the graphene. Transmission Electron Microscopy (TEM, JEOL JEM-2010) was used to observe the holes on exfoliated holey graphene sheets. Brunauer–Emmett–Teller (BET) measurements were carried out with a Micromeritics ASAP 2020 surface area and porosity analyzer to evaluate the SSA and pore distribution. To study the thermal decomposition behavior of MA and the thermal reduction process of GO, thermogravimetric analysis (TGA) was performed using a Mettler Toledo TGA/DSC 2-HT thermogravimetric analyzer. To determine the crystal structure of GO and graphene samples, X-ray diffractometry (XRD, Rigaku D/Max-2200) was performed. Moreover, the D and G bands of graphene in Raman spectroscopy were characterized with a Renishaw H27783 system. Finally, to investigate the chemical composition of the graphene samples, X-ray photoelectron spectra (XPS) were measured using a Thermo Scientific K-Alpha spectrometer. 

### 2.3. Electrochemical Measurements

A standard CR2032 coin cell structure was chosen for electrochemical studies. The electrode slurry was prepared by mixing the active material, carbon black, and polyvinylidene difluoride (PVDF, the binder) with a mass ratio of 80:10:10 in N-methyl-2-pyrrolidone (NMP). The active materials comprised activated carbon (AC) and N-doped GE or HGE in a weight ratio of 10:1. The AC was acquired from China Steel Chemical Corporation (Product number: ACS20, made from coal tar pitch) with a D_50_ particle size of 6.5 μm and a specific surface area (SSA) of 2100 m^2^/g. The slurry was uniformly mixed by an ultrasonicator and a magnetic stirrer, and then coated on a 30-μm-thick copper foil with a controlled layer thickness of 200 μm. The obtained electrode was punched into disks with a diameter of 14 mm. After vacuum drying overnight at 90 °C, two same disks divided by an NKK cellulose separator were assembled into a symmetrical supercapacitor inside an argon-filled glove box, and the electrolytes used were 1 M tetraethylammonium tetrafluoroborate (TEABF_4_) in propylene carbonate (PC) solvent. Cyclic voltammetry (CV) and galvanostatic charge-discharge (GCD) were performed using a Biologic VSP-128 potentiostat with a cell voltage range of 0–2.5 V. And electrochemical impedance spectroscopy (EIS) measurements were performed within a frequency range from 100 kHz to 10 mHz to characterize the cell’s internal impedance. The schematic illustration of the whole experiments is depicted in Figure 1.

## 3. Results and Discussion

Figure 1 shows the SEM images of N-doped graphene samples reduced at different heating rates: N-GE-15, N-GE-60, and N-HGE. The surface morphology of nitrogen-doped graphene changes significantly with the heating rate. In Figure 1a,b, N-GE-15 shows a relatively smooth and undamaged surface structure. As the heating rate was raised to 60 °C/min (N-GE-60 in Figure 1c,d, the surface became rougher and began to show holes that penetrated through the sheets. However, at this heating rate, the holes were still quite sparse. Figure 1e,f reveal that the surface of N-HGE, which was reduced with an ultra-high heating rate, has a lot more wrinkles and holes than N-GE-60. It is known that, during the thermal treatment, carbon dioxide released from the reduction of oxygen functional groups builds up an enormous pressure between the layers and results in exfoliation of graphene sheets. As the heating rate increases considerably, the pressure not only peels layers apart but also punch holes at structurally weaker spots. To more clearly reveal the holey structure, a TEM image of N-HGE is shown in Figure 2. Numerous holes can be clearly seen on the graphene sheets in the TEM image.

To evaluate the SSA and pore distribution, BET measurement results for the nanocomposites with various AC-to-holey-graphene ratios are shown in Appendix A. These figures clearly show that, although the AC-only electrode offers a larger SSA, its pores are mostly micropores (<2 nm) or mesopores with sizes of 2–4 nm. By contrast, the AC/HGE electrodes have considerably more mesopores (2–50 nm) and macropores (>50 nm) than AC, and the pore sizes increase evidently with increasing content of holey graphene.

The crystallinity of various samples was characterized by XRD. In Figure 3, MA-GO shows strong characteristic peaks at 2*θ* = 12–14°. After the heat treatment, the characteristic peaks totally disappeared and a broad and much weaker peak appeared at 2*θ* ~ 26.4°, indicating that GO had been reduced and exfoliated into few-layer graphene, and the melamine in the mixture had decomposed entirely. The TGA (thermogravimetric analysis) measurement results for melamine and MA-GO (Appendix A) also confirmed 100% decomposition of melamine at a temperature below 400 °C. As to whether the nitrogen atoms had successfully doped the graphene samples, we will examine it using XPS.

Raman spectroscopy was employed to investigate the lattice defects. The G band at 1580 cm^−1^ is related to the in-plane vibration of graphene honeycomb lattice (hexagonal lattice with a two-atom basis), whereas the D band at 1350 cm^−1^ is associated with the disorders and/or defects. The Raman spectra in Figure 4 show that all three samples have high *I*_D_/*I*_G_ ratios, indicating high defect density in the graphene lattice caused by the inserted nitrogen atoms as well as the drastic removal of oxygen functional groups during the 1000 °C post heat treatment. Many authors have reported that higher reduction temperature causes more severe damage to the graphene lattice, hence a higher *I*_D_/*I*_G_ [18,33]. It can also be seen in Figure 4 that *I*_D_/*I*_G_ increases gradually with the heating rate, which can be attributed to the increase in defect sites along the hole edges created at higher heating rate. 

The XPS analysis of nitrogen-doped graphene is shown in Figure 5. The N 1s spectra for N-GE-15, N-GE-60, and N-HGE are presented in Figure 5a, and the full spectra for all three samples are compared in Figure 5b. The deconvolution of N 1s spectra is also shown, and the peaks at 398.4, 399.8, 401.5, and 404.2 eV correspond to pyridinic, pyrrolic, graphitic, and oxidized N, respectively. Graphitic (quaternary) nitrogen is considered the most important for enhancing the electrical properties of graphene [26,27]. Pyridinic nitrogen can provide a P orbital electron to combine with the conjugated π bond of graphene to improve the electrochemical performance. Pyrrolic nitrogen provides two P orbital electrons to bond with the conjugated π bond of graphene, which facilitates electron transfer and improves capacitance. The nitrogen contents of N-GE-15, N-GE-60, and N-HGE are 3.5, 4.5, and 3.4 at.%, respectively, as listed in Table 1. Also shown in Table 1 are the carbon and oxygen atomic contents. The C 1s spectra of N-GE-60 and N-HGE are shown in Appendix A. Since the chemical composition was predominantly determined by the reduction of all oxygen functional groups at the high-temperature (1000 °C) annealing step, which is the same for all graphene samples, their C 1s spectra are quite similar, as expected. Moreover, note that N-GE-10 showed a lower oxygen content than N-GE-60 because the overall heating time for N-GE-10 (heating rate = 10 °C/min) was considerably longer than N-GE-60 (heating rate = 60 °C/min). The higher content of remaining oxygen-containing functional groups in N-GE-60 is undesirable in an organic electrolyte system, and would lead to a performance decline in the assembled SCs.

The samples were then mixed with AC (at a ratio of 1:10) to be used as the active materials in the electrode (named AC/N-GE-15, AC/N-GE-60, and AC/N-HGE). The reason we added a large percentage of AC rather than using purely graphene or holey graphene is to obtain a higher volumetric specific capacitance (F/cm^3^)—because the low tap density of graphene results in a low capacitance per volume despite its high capacitance per gram. The coin-cell supercapacitors with these electrodes and 1 M TEABF_4_/PC organic electrolyte were then assembled and tested. Figure 6a shows the CV curves measured at a potential scan rate of 50 mV/s in a voltage range of 0–2.5 V. They are all in a nearly rectangular shape, indicating a typical electric double-layer capacitor (EDLC) behavior and excellent reversibility. The specific capacitance is proportional to the area enclosed by the CV curve, and thus AC/N-HGE apparently has a much larger specific capacitance than the other two. According to the SEM results, N-HGE has a lot more holes which facilitate the electrolyte transport by offering many shortcuts for ion diffusion. Figure 6b shows the galvanostatic charge and discharge (GCD) curves for the three coin cells with a current density of 5 A/g. AC/N-HGE shows the longest constant-current charge and discharge time, indicating its largest capacitance, in agreement with the CV measurements. Figure 6c,d compare the gravimetric and volumetric specific capacitance values for the three supercapacitors at various current densities, respectively. Both kinds of specific capacitances of AC/N-HGE are considerably higher than those of AC/N-GE-60 and AC/N-GE-15 from low (0.5 A/g) to very high (50 A/g) current densities. Evidently the numerous holes of N-HGE are highly beneficial to the overall capacitance of the AC/N-HGE composite, even when the loading ratio of N-HGE to AC was only 1:10. The gravimetric and volumetric specific capacitances at 0.5 and 20 A/g of all three electrodes, along with their film densities, are listed in Table 2 for comparison.

We present the results of control experiments using only AC as the active material in the Appendix A in order to demonstrate the benefits of our AC/N-HGE composites. The large IR drop on the GCD curve (Appendix A) indicates a larger internal impedance of the AC electrode. Appendix A show that, although its specific capacitances at the lowest current density are nearly as good as those of the nanocomposites, they deteriorate much faster as the current density increases due to the slow ion diffusion into and out of the micropores (see the pore size distribution in Appendix A). We also show the performances of a device that replaces AC/N-HGE with AC/HGE (HGE is holey graphene without N-doping) in Appendix A, which clearly reveal the effects of N-doping on the SC performances by comparing the two devices. Evidently, the gravimetric and volumetric specific capacitances of AC/N-HGE are approximately twice the values of AC/HGE at low current densities. At high current densities, the ratio is even larger.

Figure 7 shows the EIS spectra of these cells. The frequency range used in this analysis was between 100 kHz and 10 mHz. The real-axis intercept on the left (i.e., the high-frequency region) represents the internal impedance, R_s_, which comprises the resistances of the active material, the electrolyte, and the contact resistance (at the interface between the active material and the collector) [34]. All curves have nearly the same R_s_ values, indicating that the overall resistances for these electrodes are similar. The diameter of the semicircle is attributed to the charge transfer resistance (R_ct_), which is related to the reaction occurring at the contact with electrolyte and is a limiting factor for the specific power of the supercapacitor [35]. Table 3 lists the R_s_ and R_ct_ values of these cells. The smallest semicircle in the EIS plot of AC/N-HGE implies the smallest R_ct_, which will result in the highest specific power. The numerous holes on N-HGE enable the ions in electrolyte to effectively diffuse into the interior of the composite, thereby reducing the impedance of ion transfer and increasing the overall specific capacitance. The extracted values of R_ct_ for AC/N-GE-15 and AC/N-GE-60 were 6.78 Ω and 8.6 Ω, respectively. These results are consistent with the oxygen content analysis in XPS (Table 1), i.e., AC/N-GE-15 exhibited a lower oxygen content than AC/N-GE-60 as a result of a longer heat treatment time due to its lower heating rate. The presence of oxygen-containing functional groups is undesirable in the organic electrolyte because it would lead to worse wettability and higher ion transport impedance, instead of offering a beneficial pseudo-capacitive effect (as in an aqueous electrolyte). Therefore, a minimized oxygen content in AC/N-GE-15 resulted in a smaller R_ct_. This effect also explains the results that AC/N-GE-15 offered slightly better overall capacitance performances than AC/N-GE-60 (see Figure 6b–d) and Table 2) although the latter showed a higher density of holes.

Next we varied the 1000 °C thermal annealing time for N-HGE to control the content of residual oxygen functional groups, which would affect the capacitor performances in organic electrolyte. Figure 8 shows the SEM images of N-HGE samples annealed for 0.5, 1, and 2 h. There is no apparent difference in the morphology of the graphene flakes and holes. This is because the exfoliation and formation of holes occurred during the ultra-rapid heating (from room temperature to 600 °C), and the subsequent reduction reaction of oxygen functional groups at 1000 °C was very mild. 

Figure 9 shows the Raman spectra for these N-HGE samples treated with different annealing times. The *I*_D_/*I*_G_ ratio increases slightly with the annealing time, indicating an increase in defect sites due to the reduction of oxygen functional groups at 1000 °C. 

The XPS analysis of the three N-HGE samples is shown in Figure 10. The N 1s spectra for N-HGE-0.5hr, N-HGE-1hr, and N-HGE-2hr are presented in Figure 10a, and the full spectra for all three samples are compared in Figure 10b. The carbon, nitrogen, and oxygen atomic contents for these samples are compared in Table 4. As expected, the oxygen content decreased with the annealing time (from 4.3% for N-HGE-0.5hr to 3.8% for N-HGE-2hr), confirming that the oxygen-containing functional groups can be effectively removed by high-temperature annealing. However, the N content also dropped from 3.6% to 2.9% in the meantime.

Figure 11a shows the CV curves (at a scan rate of 50 mV/s) for the supercapacitors made with the composites of AC and the three N-HGE samples annealed for different times. All the curves are nearly rectangular, and clearly the enclosed area, which represents the capacitance, increases with increasing annealing time. In Figure 11b, the GCD curves (at 5 A/g) for the three coin cells also reveal that the capacitance increases with the annealing time. When the thermal annealing time at 1000 °C is increased, more oxygen-containing functional groups on N-HGE are removed. In an organic electrolyte, this results in a better wettability and improved ion transport, which increase the utilization of specific area and reduce the internal resistance. Figure 11c,d display the gravimetric and volumetric specific capacitances vs. current density, respectively. They clearly show that, at all current densities, both kinds of specific capacitances improved as the 1000 °C annealing time was increased. The gravimetric and volumetric specific capacitances at 0.5 and 20 A/g of these three electrodes, along with their film densities, are listed in Table 5 for comparison. As the annealing time increased from 0.5 h to 2 h, both the gravimetric and volumetric specific capacitances were improved by 36% at 0.5-A/g charging rate, and at 20-A/g charging rate the improvement soared to more than 46%. 

Figure 12 shows the EIS spectra of these three SCs. The impedance will affect the capacitance under fast charging and discharging conditions. The resistance values extracted from circuit fitting are listed in Table 6. As the annealing time increases from 0.5 h to 2 h, R_ct_ decreases monotonically from 6.72 Ω to 4.19 Ω. This is because of the removal of excess oxygen-containing functional groups, which results in better wetting of the N-HGE in the organic electrolyte, thereby improving the electrical conductivity and the electrochemical activity of the AC/N-HGE composite. 

The energy density and power density performances of these SCs can be evaluated with the Ragone plot. As shown in Figure 13a,b, the gravimetric and volumetric energy densities of AC/HGE-2hr are greater than the others at all power density levels. At the lowest discharging rate, the energy density reaches 33 Wh/kg (or 10.6 Wh/L). Most importantly, at the highest output power density of 31,000 W/kg (or 10,000 W/L), its energy density is still maintained at 11 Wh/kg (or 3.5 Wh/L). 

Though it is difficult to make direct comparison with other methods, we provide a Ragone plot (Appendix A) comparing the specific energy vs. specific power performances of our results and other benchmark studies using similar materials [11,18,36,37,38,39] (Appendix A). Our results outperform most of them, and are comparable to the best works among them. In a practical sense, our method offers a simple, fast, and low-cost process to prepare the N-doped holey graphene and achieve considerable improvements on supercapacitor performances. Additionally, since AC is the most common active material in commercial SCs currently, our method can be a valuable solution for the industry because the N-HGE produced by this simple process can be widely used as an effective performance booster at a very small content (1/10 of the AC).

## 4. Conclusions

N-doped holey graphene was synthesized by a cost-effective ultra-rapid heating approach using melamine as the N source, and organic-electrolyte supercapacitors using the composite of AC and N-HGE were fabricated. It was found that holey graphene can effectively enhance the performances of the SCs compared to the non-holey counterparts owing to the shortened ion diffusion paths. Furthermore, N doping can improve the wettability and conductivity of HGE, thus increasing the electrochemical activity and specific capacitance of the composite. However, the residual oxygen functional groups on N-HGE has unfavorable effects in the organic electrolyte. Increasing the annealing time at 1000 °C from 0.5 h to 2 h improved the specific capacitance by 36% at 0.5-A/g current density, and more than 46% at 20 A/g. The SC with the AC/N-HGE-2hr electrodes exhibited the best specific energy at all power density levels, and retain a high specific energy of 11 Wh/kg (or 3.5 Wh/L) at the largest power density of 31,000 W/kg (or 10,000 W/L). We believe that the AC/N-HGE-2hr composite prepared by our simple and low-cost method has great potential for high-performance energy storage applications.

## Data Availability

The data presented in this study are available on request from the corresponding authors.

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
