# Peer review of "Study on the Application of Nitrogen-Doped Holey Graphene in Supercapacitors with Organic Electrolyte"

_nanomaterials, 2023, doi:10.3390/nano13101640_

Round 1

Reviewer 1 Report

In this work, the authors use an easy and simple route to create N-doped graphene (N-HGE) from melamine with graphite oxide (GO) and used a rapid heating method to create the N-HGE material that had better performance in supercapacitors. This study may find importance in the field of electrochemical energy storage. However, there are still some major concerns in the manuscript that need to be addressed to be suitable for publication.

 1.     The introduction needs a major revamp. Please discuss the different between holey graphene and graphene with 3D porous structure, and their implications on the electrochemical performance.

 2.     Some relevant work should be compared and discussed: https://www.nature.com/articles/ncomms5554; https://pubs.acs.org/doi/full/10.1021/acsanm.0c02781; https://pubs.acs.org/doi/full/10.1021/nn402077v; https://www.sciencedirect.com/science/article/pii/S0025540817306645. 

 3.     “Holey” or “porous” graphene? The authors claim to have created N-HGE with “numerous holes all over the graphene sheets”, however, the product is not adequately characterized.

BET should be carried out to analyse the pore size. Only SEM images with relatively low resolution (~1µm scale bar), which appear to be porous graphene got pressed down (during SEM sample prep) rather than holey graphene.

Please provide SEM (or preferably TEM) of the N-HGE on a flat surface (exfoliate and then filter on a membrane, or deposit on the TEM grid) to confirm the product is actually holey graphene as claimed.

 4.     The electrode is prepared by mixing AC and N-HGE at weight ratio of 10:1. So the dominant electrochemical characteristics are derived from AC. Control experiments using AC only and N-HGE only should be provided.

 5.     For N1s comparison (Fig 4a-c and 9a-c), it is advised to stack them up into a single figure using layers for easier comparison. See https://pubs.acs.org/doi/10.1021/acsanm.2c04306 (Fig 1d,e) as an example.

 6.     As carbon-based materials, it is essential to compare the C1s spectra of the produced graphene.

 7.     For reporting volumetric capacitances, please use F/cm3 instead of F/mL (Fig 5d and 10d).

 8.     A Ragone plot comparing this work with others should be presented.

Reviewer 2 Report

This paper deals with “Study on the application of nitrogen-doped holey graphene in supercapacitors with organic electrolyte". However, the representative preparation(schematic illustration, including the yields of prepared materials and their characterization, should be described in details (SEM, TEM, TGA, XRD, …). And authors should compare with other holey graphene based electrode systems (or reported previously) and provide the academic novelty or technical advantages.

Round 2

Reviewer 1 Report

Thanks the authors for their effort in revising the manuscript. It is now suitable for publication.

Author Response

Thanks for the review comments to make the article better.